# Facile access to bicyclo[2.1.1]hexanes by Lewis acid-catalyzed formal cycloaddition between silyl enol ethers and bicyclo[1.1.0] butanes

Sai Hu[1,2,3,4], Yuming Pan[2,3,4], Dongshun Ni [2,3] ✉ & Li Deng [2,3] ✉

Saturated three-dimensional carbocycles have gained increasing prominence in synthetic and medicinal chemistry. In particular, bicyclo[2.1.1]hexanes (BCHs) have been identified as the molecular replacement for benzenes. Here, we present facile access to a variety of BCHs via a stepwise two-electron formal (3 + 2) cycloaddition between silyl enol ethers and bicyclo[1.1.0]butanes (BCBs) under Lewis acid catalysis. The reaction features wide functional group tolerance for silyl enol ethers, allowing the efficient construction of two vicinal quaternary carbon centers and a silyl-protected tertiary alcohol unit in a streamlined fashion. Interestingly, the reaction with conjugated silyl dienol ethers can provide access to bicyclo[4.1.1]octanes (BCOs) equipped with silyl enol ethers that facilitate further transformation. The utilities of this methodology are demonstrated by the late-stage modification of natural products, transformations of tertiary alcohol units on bicyclo[2.1.1]hexane frameworks, and derivatization of silyl enol ethers on bicyclo[4.1.1]octanes, delivering functionalized bicycles that are traditionally inaccessible.

The strategic replacement of benzene with conformationally rigid and stable C(sp³)-enriched polycyclic scaffolds in small molecules represents an emerging trend in medicinal chemistry. Attributed to their constrained geometries and precisely oriented pendant substituents, these saturated polycycles effectively emulate the topological characterisics of substituted benzenes, which allows for the preservation of desired interactions with biomacromolecules while enhancing the pharmacokinetics, solubility, and metabolic stability of drug candidates[1–5]. Recent studies have identified 1,2-disubstituted bicyclo[2.1.1]hexanes as potential bioisosteres for *ortho*-disubstituted benzenes with retained biological activity validated by in vitro experiments (Fig. 1a)[6,7]. Hence, there is an increasing demand for development of efficient strategies for streamlined access to these bicycles[6–32]. One of most common methods to construct BCH skeleton is by an intramolecular [2 + 2] cycloaddition of 1,5-diene under the irradiation of light[6–14]. Alternatively, an intermolecular cycloaddition between bicyclo[1.1.0]butanes (BCBs) and alkenes is highly desirable since it allows the efficient construction of bicyclic ring through the fusion of two readily available starting materials. Pioneering studies were disclosed by Blanchard[16] in 1966 and De Meijere[17] in 1986. Subsequently, Wipf group reported an intramolecular variant of this cycloaddition under thermal condition in 2006[18].

More recently, by taking advantage of the ready availability and inherent ring strain of BCBs[20,33], the exploration of new strategies to the cycloaddition between BCBs and alkenes in the generation of various BCHs has attracted intensive attentions[19–32,34–39]. According to the reported reaction processes, most methods could be categorized into two modes: 1) radical pathway; and 2) two-electron pathway

[1]Department of Chemistry, Zhejiang University, Hangzhou, China. [2]Key Laboratory of Precise Synthesis of Functional Molecules of Zhejiang Province, School of Science and Research Center for Industries of the Future, Westlake University, Hangzhou, China. [3]Institute of Natural Sciences, Westlake Institute for Advanced Study, Hangzhou, China. [4]These authors contributed equally: Sai Hu, Yuming Pan. ✉e-mail: nidongshun@westlake.edu.cn; dengli@westlake.edu.cn

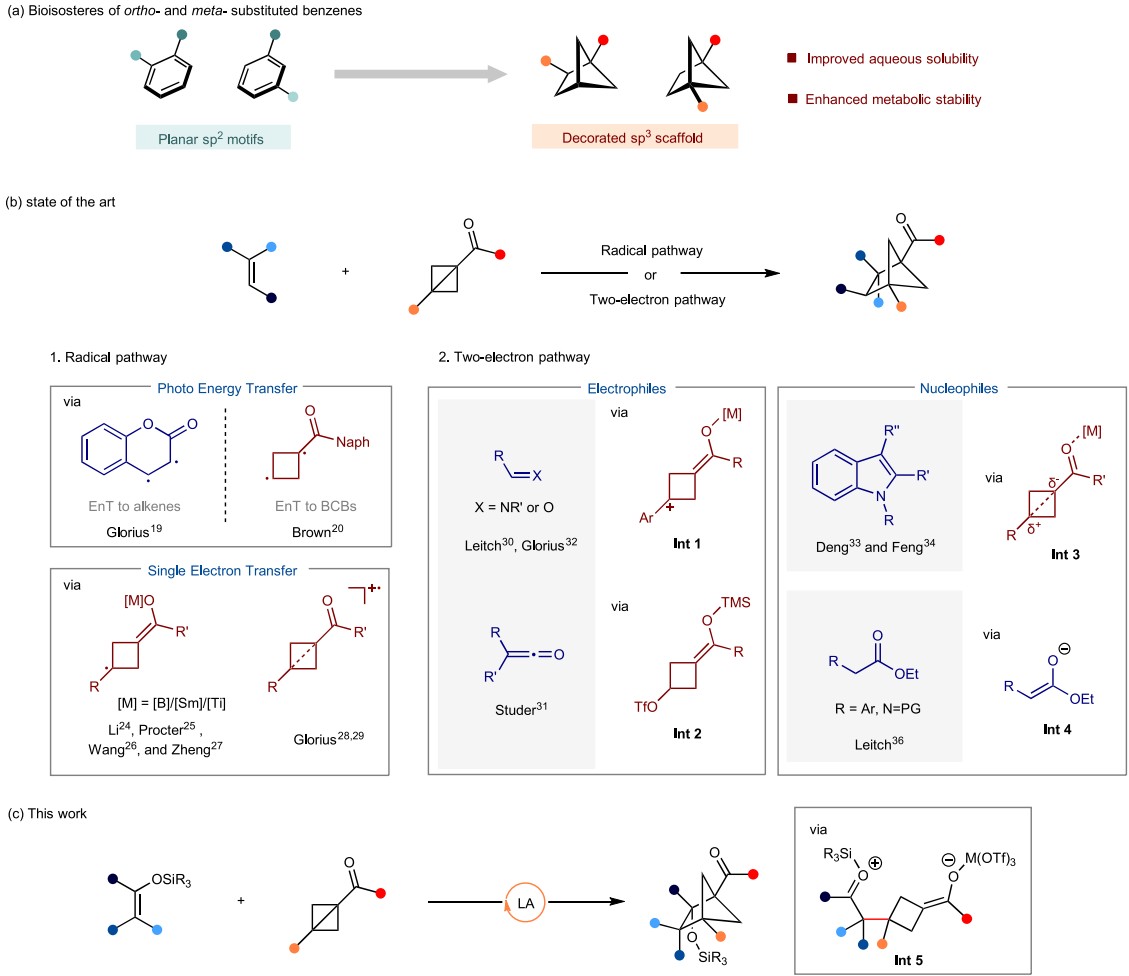

**Fig. 1 | Importance of bicyclo[2.1.1]hexanes (BCHs) and synthetic strategies by formal cycloaddition between bicyclo[1.1.0]butanes (BCBs) and alkenes.**
**a** BCHs behave as bioisosteres of *ortho*- and *meta*-substituted benzenes. **b** State of

the art in the cycloaddition of BCBs and alkenes to access BCHs. **c** Cycloaddition of BCBs and silyl enol ethers to access BCHs.

(Fig. 1b). By utilizing the photoinduced energy transfer strategy, Glorius[19] and Brown[20] groups respectively described elegant cycloaddition of BCBs and alkenes toward bicyclo[2.1.1]hexanes. The reaction was initiated by the excitation of either alkene or BCB to generate a diradical intermediate. Subsequently, Bach and coworkers achieved an enantioselective cycloaddition of 2(1*H*)-quinolones and BCBs with a chiral mediator[22]. Recently, Jiang and coworkers developed a highly enantioselective cycloaddition of vinylazaarenes and BCBs under photosensitized chiral phosphoric acid catalysis[23]. Li[24] and Wang[26] groups developed a boryl-pyridine catalytic system to activate BCB as a cyclobutyl radical intermediate. Meanwhile, Procter group applied SmI$_2$ as a single electron reductant to achieve the insertion of electron-deficient alkene into BCB[25]. Very recently, Zheng group described a Ti-catalyzed formal cycloaddition of BCB and 2-azadienes to synthesize aminobicyclo[2.1.1]hexanes[27]. Lately, Glorius accomplished the coupling of phenol and BCB by leveraging a photoredox process, which has been applied in the formal cycloaddition of non-activated alkenes (Fig. 1b, 1)[28,29]. All the reactions above entailed the generation of radical species, which limited the substrate scope.

Bicyclo[1.1.0]butanes could be activated as an enolate nucleophile upon central σ-bond cleavage mediated by Lewis acid to attack electrophilic reagents such as aryl aldimine by Leitch group[30], aldehyde by Glorius group[32], or ketene by Studer group[31], followed by the intramolecular cyclization to complete the formal cycloadditions. We demonstrated that Lewis acid could activate BCBs as electrophiles to react with indoles as the nucleophiles to construct complex indoline

polycycles (Figs. 1b, 2)[34]. Notably, if a wide variety of nucleophiles could be utilized, this approach could be developed into a versatile strategy that complements existing methods involving radical or ionic intermediates to access BCHs.

More recently, Feng reported the use of silver triflate to promote reactions of BCBs and indoles but with opposite regioselectivity[35,36]. Very recently, Leitch group described a formal cycloaddition of BCBs via an enolate intermediate **Int 4** by treatment of glycine imine or arylacetate derivatives with stoichiometric amount of LHMDS[37]. Despite these successful examples, there is a huge and unexplored chemical space for these bicycles, and a more general strategy to expediently construct such moieties with simple alkenes is highly demanded.

Here, we envisaged that the silyl enol ether, as the nucleophile, would be a suitable candidate in this scenario due to its importance in different cycloaddition reactions[40–48]. It is noteworthy that silyl enol ethers could be easily prepared by a one-step silylation of simple ketone motifs, which are presented in numerous compounds and widely used in synthetic chemistry. Although silyl enol ethers exhibit robust nucleophilicity in various synthetic contexts, such as Mukaiyama-aldol reaction and Michael addition, their nucleophilicity towards the BCB remains unexplored[37,39]. Furthermore, the subsequent intramolecular aldol-type cyclization presents a challenge due to the formation of sterically hindered vicinal quaternary carbon centers. Despite these concerns, we aim to explore the formal (3 + 2) cycloaddition

**Table 1 | Optimization of reaction conditions**[a]

| Entry | Lewis acid | Solvent | Results[b] |
|---|---|---|---|
| 1 | Cu(OTf)$_2$ | DCM | **3a** (5%), **4** (45%) |
| 2 | Zn(OTf)$_2$ | DCM | **3a** (14%), **4** (57%) |
| 3 | Ni(OTf)$_2$ | DCM | **3a** (3%), **4**(45%) |
| 4 | AgOTf | DCM | **3a** (6%), **4** (46%) |
| 5 | Sc(OTf)$_3$ | DCM | **3a**(56%) |
| 6 | B(C$_6$F$_5$)$_3$ | DCM | **4** (43%) |
| 7 | Eu(OTf)$_3$ | DCM | **3a** (84%) |
| 8 | Gd(OTf)$_3$ | DCM | **3a** (74%) |
| 9 | Tm(OTf)$_3$ | DCM | **3a** (92%) |
| 10 | Lu(OTf)$_3$ | DCM | **3a** (90%) |
| 11 | Yb(OTf)$_3$ | DCM | **3a** (97%)(96%)[c] |
| 12 | YbCl$_3$ | DCM | n.r. |
| 13 | Yb(OAc)$_3$ | DCM | n.r. |
| 14 | Yb(OTf)$_3$ | toluene | **3a** (98%)(98%)[c] |
| 15 | Yb(OTf)$_3$ | THF | **3a** (42%) |
| 16 | Yb(OTf)$_3$ | MeCN | **3a** (63%) |
| 17 | - | DCM | n.r. |

[a]Reactions were run on a 0.10 mmol scale, with 0.10 mmol silyl enol ethers **1a**, 0.13 mmol BCB **2a**, and 0.01 mmol Lewis acid.
[b]Yields were determined by 1H NMR analysis of the unpurified reaction mixture with 1,1,2,2-tetrachloroethane or 1,3,5-trimethoxybenzene as an internal standard.
[c]Yield of the reaction with 5 mol% of Yb(OTf)$_3$.

between silyl enol ether and BCB to realize one-step access to a variety of BCH frameworks (Fig. 1c).

## Results and discussion

### Reaction optimization

With these considerations in mind, we began our investigations with screening studies of Lewis acids on their abilities to promote the reaction of triisopropylsilyl phenyl enol ether (**1a**) from acetophenone and naphthyl BCB (**2a**). In the presence of late transition metal-derived triflate salts, only trace amount of desired cycloadduct **3a** was observed by NMR analysis of the crude reaction mixture (Table 1, entries 1–4). The major product was identified to be cyclobutyl silyl enol ether **4**, presumably derived from nucleophilic addition followed by silyl migration.

Additionally, B(C$_6$F$_5$)$_3$ would only give byproduct **4** (Table 1, entry 6). On the other hand, we found that with lanthanide triflates such as Eu(OTf)$_3$, Gd(OTf)$_3$, Tm(OTf)$_3$, Lu(OTf)$_3$ and Yb(OTf)$_3$ the reaction afforded the desired product **3a** in high yield (Table 1, entries 7–11). Among them, Yb(OTf)$_3$ was the optimal catalyst, allowing the reaction to proceed in 97% yield (Table 1, entry 11). Interestingly, switching the counterion from triflate to chloride or acetate is detrimental to this transformation, illustrating the acidity of Lewis acid is the key to success. (Table 1, entries 12–13). Further optimization studies identified toluene as another suitable solvent (Table 1, entries 14–16). Pleasantly, the catalyst loading could be decreased to 5 mol% without negative impact (Table 1, entries 11 and 14). Control experiment illustrated that no reaction occurred in the absence of Lewis acid (Table 1, entry 17).

### Substrate scope

With the optimal conditions in hand, the generality of substrates in this cycloaddition was investigated as depicted in Fig. 2. The reactions of BCB **2a** and various silyl enol ethers **1a**–**d** derived from acetophenones were examined, which proceeded to completion, affording the desired products **3a**–**d** in 45–93% yields. Within these substrates, those bearing more stable silyl groups resulted in higher yields. The reactions tolerated a wide variety of aromatics with substituents on various positions and with different electronic properties (**3e**–**p**). 1-Naphthyl and 2-naphthyl substituted silyl enol ethers worked well to form the corresponding products (**3q** and **3r**). The structure of cycloadduct **3r** was unambiguously confirmed by single crystal X-ray diffraction analysis.

Heterocycles commonly used in medicinal chemistry such as furan and thiophene were compatible, giving their corresponding bicycles (**3s** and **3t**) in good yields. Additionally, silyl enol ethers bearing alkenyl and alkynyl groups were readily converted into the corresponding cycloadducts (**3u** and **3v**) in high yields. Importantly, the reactions with aliphatic substituted silyl enol ethers with pendant functional groups such as chloride (**3y**), silyl ether (**3z**), olefin (**3aa**), electron-rich arene (**3ab**), and cyclopropane (**3ac**) performed well to furnish the corresponding cycloadducts in 47–90% yields.

Next, a series of cyclic silyl enol ethers derived from 5-, 6-, 7-membered cyclic ketones including those fused with aromatic and heteroaromatic rings were assessed, producing the corresponding assortment of bridged polycycles (**3ad**–**an**) in synthetically useful yields. Interestingly, these reactions could convert cyclic silyl dienol ethers to tricyclic alkenes **3al** and **3am**, which are suitable for further elaborations to synthesize more complex molecular frameworks.

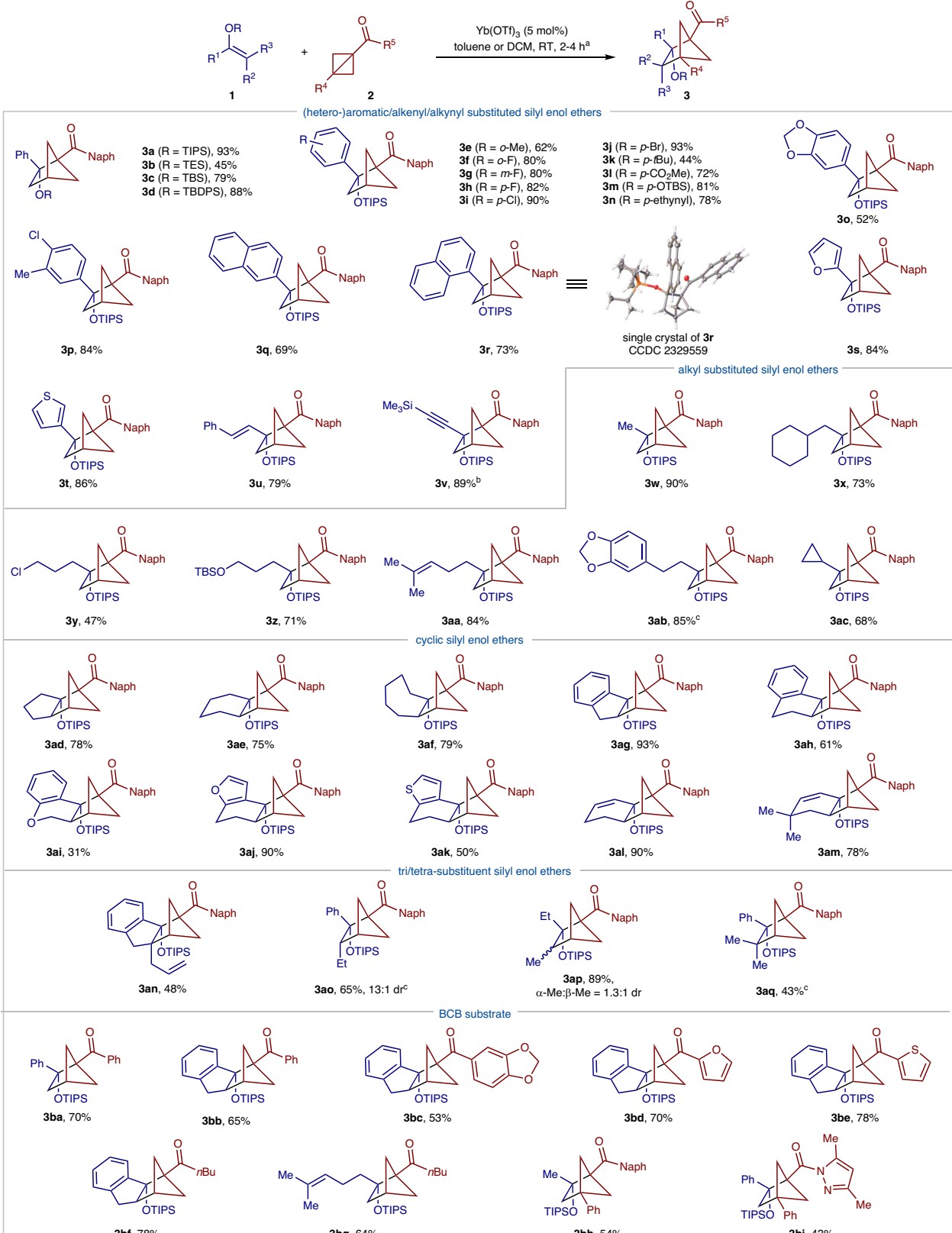

**Fig. 2 | Substate scope of silyl enol ethers and BCBs.** [a]Reaction conditions: unless indicated otherwise, the reaction of silyl enol ether 1 (0.2 mmol), and BCB 2 (0.26 mmol) was carried out in DCM (2 mL) or toluene (2 mL) in the presence of Yb(OTf)$_3$ (0.01 mmol) at room temperature for 2–4 h. The yield was of isolated and purified products. [b]6 h. [c]12 h. β-Me, methyl group *syn* to ethyl group; α-Me, methyl group *anti* to ethyl group. Naph 2-naphthyl, TIPS triisopropylsilyl, TES triethylsilyl, TBS *tert*-butyldimethylsilyl, TBDPS *tert*-butyldiphenylsilyl, *o* ortho, *m* meta, *p* para.

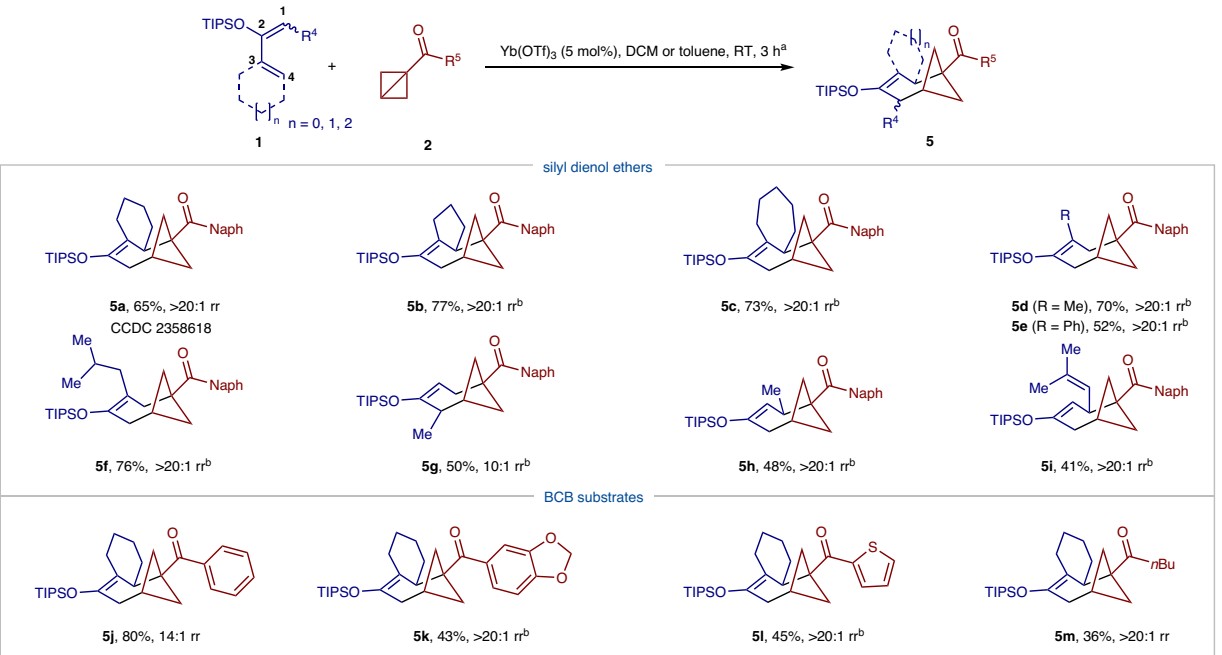

**Fig. 3 | Substate scope of silyl dienol ethers and BCBs.** [a]Reaction conditions: unless indicated otherwise, the reaction of silyl dienol ether **1** (0.2 mmol), and BCB **2** (0.26 mmol) was carried out in DCM (2 mL) or toluene (2 mL) in the presence of Yb(OTf)$_3$ (0.01 mmol) at room temperature for 3 h. The yield was of isolated and purified products. [b]The reaction was run in the presence of Sc(OTf)$_3$ (0.02 mmol) at room temperature for 30 min. rr: regioselectivity ratio of (4 + 3) to (3 + 2) cycloadduct. Naph 2-naphthyl, TIPS triisopropylsilyl.

Notably, acyclic tri- and tetra-substituted silyl enol ethers are viable substrates, allowing the rapid construction of highly substituted and compact bicycles **3ao–aq**. It is worth noting that the reaction with different trisubstituted silyl enol ethers resulted in distinct diastereoselectivities. In particular, **3ao** were obtained in high diastereoselectivity (13:1 dr) from the corresponding silyl enol ether **1ao** (Z/E = 10:1) derived from 1-phenyl-1-butanone, while **3ap** was formed as a mixture of diastereomers (1.3:1 dr) with silyl enol ether **1ap** (Z/E = 7.4:1) derived from 3-pentanone (Supplementary Figs. 4–8).

Finally, we turned our attention to the evaluation of BCB scope. BCBs with aromatic, heteroaromatic and aliphatic substituents such as benzene (**3ba–bb**), 1,2-methylenedioxybenzene (**3bc**), furan (**3bd**), thiophene (**3be**), and n-butyl (**3bf–bg**) are well tolerated in current conditions. To our delight, the reaction with 3-substituted BCB **2g** and **2h** also proceeded smoothly to give the cycloadducts bearing three quaternary carbon stereocenters (**3bh** and **3bi**). The ability to tolerate variations of both silyl enol ethers and BCBs indicated the potential of this method to generate a wide range of BCHs that are not easily accessible by previously reported methods.

Interestingly, structurally intriguing bicyclo[4.1.1]octane (BCO) architecture **5a** was formed in the reaction of cyclohexenyl silyl dienol ether and BCB **2a** under standard conditions (Fig. 3)[49]. Grygorenko et al. analyzed the structural characteristics of BCOs and suggested that they could serve as novel isosteres for multi-substituted benzenes[50]. However, regioselectivity issues between formal (4 + 3) and (3 + 2) cycloaddition might occur in this reaction. Further optimization studies identified Sc(OTf)$_3$ as an alternative efficient Lewis acid that promotes formal (4 + 3) cycloaddition. With both Sc(OTf)$_3$ and Yb(OTf)$_3$ catalysts, we examined the generality of formal (4 + 3) cycloaddition with both reaction components. The reaction conditions were compatible with silyl dienol ether containing 5 and 7-membered rings, affording tricycles **5b** and **5c** bearing bicyclo[4.1.1] octane units. Silyl dienol ethers derived from aliphatic and aromatic vinyl ketones gave good yields (**5d–g**). Notably, silyl trienol ether also underwent the formal (4 + 3) cycloaddition selectively in synthetically useful yield (**5i**). Different substituents on BCBs such as benzene (**5j**), 1,2-methylenedioxybenzene (**5k**), thiophene (**5l**), and n-butyl (**5 m**) groups are well tolerated, delivering desired (4 + 3) cycloadducts in medium to good yields. The divergent synthesis of both bicyclo[2.1.1] hexanes and bicyclo[4.1.1]octanes demonstrated the utility of silyl enol ethers and their derivatives in the construction of different bicycles.

## Synthetic applications

To illustrate the utilities of this method, we conducted this reaction on a preparative scale with silyl enol ether **1w** and BCB **2a**, which afforded the cycloadduct **3w** in 91% yield (770 mg) (Fig. 4a). Both the ketone and tertiary alcohol moieties in **3w** provided handles for further synthetic elaborations to procure a wide array of new bicyclo[2.1.1]hexanes (Fig. 4b). Specifically, subjecting bicycle **3w** to the Wittig reaction condition led to olefin **6**. After deprotection of the silyl group, the free alcohol **7** was proved versatile for various transformations, including the dehydration to olefin **8** by treatment with Burgess reagent and fluorination to **9** in the presence of diethylaminosulfur trifluoride (DAST) reagent.

Furthermore, isonitrile **10** would be readily obtained using Shenvi's methodology[51]. All of derivatizations above further extended the attainability of new bicyclo[2.1.1]hexanes using this developed manifold. Additionally, the resulting silyl enol ether unit from the (4 + 3) cycloadducts could be further transformed to ketone **11** via desilylation and α-hydroxyketone **12** via dihydroxylation. Importantly, owing to its generality with respect to silyl enol ethers, this method could be applied to late-stage modifications of natural products, as bridged polycycles **13** and **14** were formed in synthetically useful yields from commercially available dihydro-β-ionone and cholestenone through ketone silylation and formal cycloaddition sequence (Fig. 4c).

To gain more insights into the reaction mechanism, we conducted ¹³C kinetic isotope effect experiments by employing Singleton's method[52–54] with natural abundance of ¹³C. We found only carbon c showed a pronounced kinetic isotope effect, which suggested the reaction possibly underwent a stepwise process instead of a concerted

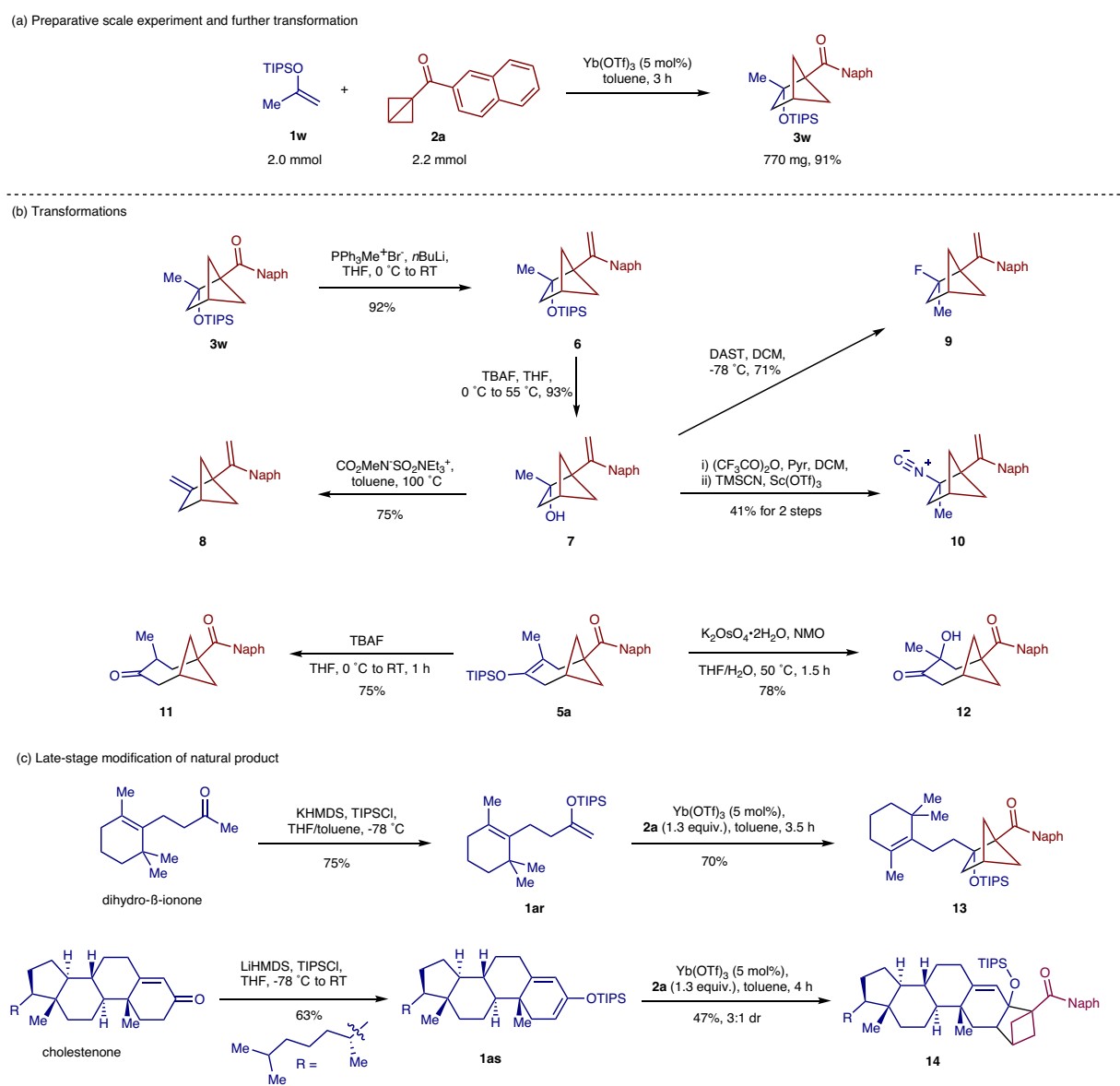

**Fig. 4 | Applications. a** Scale-up synthesis of **3w**. **b** Derivatization of **3w** and **5a**. **c** Late-stage modification of natural products. Naph 2-naphthyl, TIPS triisopropylsilyl, TBAF tetrabutylammonium fluoride, DAST diethylaminosulfur trifluoride, Pyr pyridine, NMO 4-methyl morpholine N-oxide, HMDS bis(trimethylsilyl)amide.

pathway (Fig. 5a). Therefore, a possible mechanism was proposed and illustrated in Fig. 5b. The reaction started from the nucleophilic addition of silyl enol ether with Lewis acid-activated BCB. The formed zwitter-ionic intermediate **Int 5** then underwent an intramolecular aldol reaction to give the cycloadduct **3**. Otherwise, silyl migration might occur after the in situ generation of **Int 5**, which leads to the cyclobutyl silyl enol ether **4**. Additionally, when BCB reacts with silyl dienol ether to generate **Int 5′**, the reaction could proceed through 1,4-addition, delivering bicyclo[4.1.1]octane **5**. The regioselectivity of formal (3 + 2) and (4 + 3) cycloadditions with silyl dienol ethers appears to be substrate-dependent. As demonstrated by the formation of (3 + 2) and (4 + 3) cycloadducts (**3u, 3al, 3am, 5a-5h**), the steric and electronic properties of the substituents on silyl dienol ethers have a significant influence on which pathway is favored. Considering the proposed reaction process, the origin of distinction in diastereoselectivity possibly arose from the steric bulk of R-substituent geminal to silyloxyl group on silyl enol ether **1** in the aldol-type cyclization step, which was elucidated by the proposed Felkin-Anh type model (Supplementary Fig. 9).

In summary, we developed a Lewis acid-catalyzed formal (3 + 2) cycloaddition of silyl enol ethers and BCBs[55]. This reaction exhibits

high efficiency along with mild condition, operative simplicity, and broad substrate scope with respect to silyl enol ethers, which are readily available from ketone precursors. Notably, ketone is one of the most widely presented functional groups in organic molecules including natural products. Moreover, this formal (3 + 2) cycloaddition can be further extended to a (4 + 3) variant with silyl dienol ethers. This discovery enables the divergent syntheses of different structurally intriguing polycyclic frameworks and highlights the great potential of silyl enol ethers in the synthesis of polycycles. Importantly, this method should provide a broadly useful entry into bicyclo[2.1.1]hexanes from easily accessible starting materials covering a wide range of structural complexity. Further studies for the development of new reactions involving other widely available starting materials and BCBs are ongoing in our laboratory.

## Methods
### General procedure for the formal cycloaddition of silyl (di)enol ethers and bicyclo[1.1.0]butanes
A 1-dram screw-cap vial with a stir bar was charged with silyl (di)enol ether **1** (0.2 mmol), BCB **2** (0.26 mmol) and Yb(OTf)₃ (0.01 mmol).

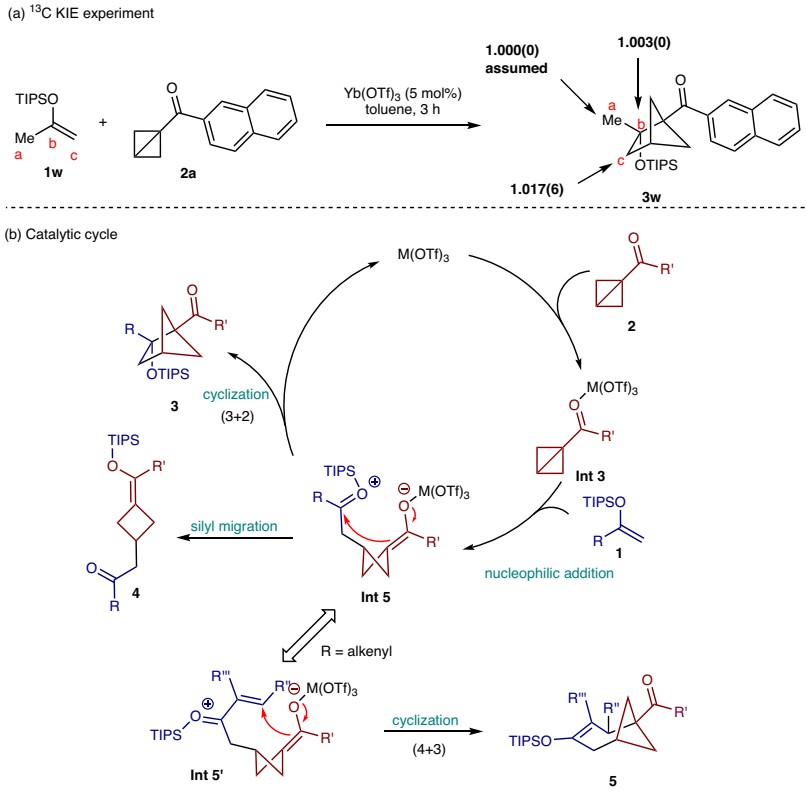

**Fig. 5 | Proposed mechanism. a** [13]C kinetic isotope effect experiment. **b** Catalytic cycle.

Then the vial was sealed with a septum, evacuated, and backfilled with $N_2$ (×3). Under $N_2$ atmosphere, dry DCM or toluene (2 mL) was added rapidly. The septum was quickly replaced with a screw cap. The reaction was stirred for 2–4 h until silyl enol ether was completely consumed. The reaction mixture was filtered through a pad of silica-gel and washed with DCM. The filtrate was concentrated on rotary evaporator and the residue was directly purified by silica-gel flash column chromatography to afford the desired product **3** or **5**.

### General procedure for the formal cycloaddition of silyl dienol ethers and bicyclo[1.1.0]butanes

A 1-dram screw-cap vial with a stir bar was charged with silyl dienol ether **1** (0.2 mmol), BCB **2** (0.26 mmol). Then the vial was sealed with a septum, evacuated, and backfilled with $N_2$ (×3). Under $N_2$ atmosphere, dry DCM or toluene (2 mL) was added rapidly. When fully dissolved, $Sc(OTf)_3$ (0.02 mmol) was added in one portion, and the septum was quickly replaced with a screw cap. The reaction was stirred for 30 min until silyl dienol ether was completely consumed. After quenching with triethylamine (0.2 mL), the reaction mixture was filtered through a pad of deactivated silica-gel and washed with DCM. The filtrate was concentrated on rotary evaporator and the residue was directly purified by silica-gel flash column chromatography to afford the product **5**.

### Data availability

The details of experimental procedures and the data about the findings of this study are available within the article and its supplementary information. Crystallographic data for the structures reported in this article have been deposited at the Cambridge Crystallographic Data Centre, under deposition numbers CCDC 2329559 (**3r**) and 2358618 (**5a**). Copies of the data can be obtained free of charge via https://www.ccdc.cam.ac.uk/structures/. All data are available from the corresponding author upon request.

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

## Acknowledgements

We are grateful for financial support from the National Natural Science Foundation of China (grant no. 22201233 to D.N., grant no. U22A20389 to L.D.), and the Foundation of Westlake University; the Leading Innovative and Entrepreneur Team Introduction Program of Zhejiang (2020R01004). We thank the Instrumentation and Service Center for Molecular Sciences and Physical Sciences at Westlake University for the assistance in measurement/data interpretation. We also thank Dr. Xiaohuo Shi at Westlake University for his assistance in the measurement of NMR and Dr. Yinjuan Chen at Westlake University for her assistance in the measurement of HRMS, Dr. Fucheng Leng at Westlake University for assistance with X-ray measurement.

## Author contributions

S.H., Y.P. and D.N. carried out the experiments and data analysis work. D.N. and L.D. designed the reaction and directed the project. The paper was written by D.N. and L.D. All authors contributed to discussions. S.H. and Y.P. contributed equally.

## Competing interests

The authors declare no competing interests.
