## [Peer Review File · Nature Communications]

Facile access to bicyclo[2.1.1]hexanes by Lewis acid-catalyzed formal cycloaddition between silyl enol ethers and bicyclo[1.1.0]butanesREVIEWER COMMENTS

Reviewer #1 (Remarks to the Author):

The chemistry of bicyclobutanes (BCBs) is currently of broad interest due to their ability to provide access to sp³-rich structures that are highly sought after in drug discovery. In this manuscript, Ni, Deng, and their co-workers describe an efficient Lewis-catalyzed approach for constructing multisubstituted valuable bicyclo[2.1.1]hexanes (BCHs) and bicyclo[4.1.1]octanes (BCOs) using bicyclobutanes and silyl enol ethers as the key starting materials. The reaction exhibits a wide functional group tolerance for silyl enol ethers and high yields in most cases. This work makes a significant contribution to BCB chemistry. I fully support publication of this impressive work in Nature Communications after the revisions below have been taken into consideration.

1-The Leitch group pioneered the design of Lewis acid-catalyzed (3+2) cycloaddition of bicyclo[1.1.0]butanes (BCBs) in 2022. The design of Lewis acid-catalysed (3+2) cycloaddition of bicyclo[1.1.0]butanes (BCBs), including the cycloaddition of BCBs with electron-enriched alkenes (ref. 22 and 25), is general, as evidenced by the rapidly increasing number of reports cited by the authors (ref. 27-32). Some (3+2) cycloadditions of BCBs are now even enantioselective (J. Am. Chem. Soc. 2023, 145, 24466–24470; J. Am. Chem. Soc. 2024, 146, 8372–8380). However, the (4+3) cycloadducts, BCOs, are intriguing and significant, as they could serve as novel isosteres for multisubstituted benzenes, as reported by Grygorenko (Chem. Eur. J. 2023, e202303859). The present Lewis acid-catalyzed cycloaddition of BCBs with dienes is a unique (4+3) cycloaddition, offering a catalytic approach to obtain valuable BCOs. Consequently, the aforementioned significant studies should be referenced, the introduction requires revision, and the substrate scope of the (4+3) reaction needs to be expanded. Furthermore, the authors should cite recent progress in (hetero)-(4+3) cycloadditions of BCBs (such as ChemRxiv, DOI: 10.26434/chemrxiv-2024-r9c2g)

2- The authors suggested that enolate addition to BCBs is the reaction mechanism. However, this process is not frequently observed in BCB chemistry. The initial attack by the carbanion, is found in recent work from Leitch and co-workers on enolate additions to bicyclobutanes that is not cited (Chem. Commun., 2023, 59, 13847–13850).

3- It is disappointing that the 1,3-disubstituted BCBs cannot be used in the (3+2) and (4+3) cycloadditions of BCBs. Have any other byproducts been observed in the reaction of these 1,3-disubstituted BCBs? This should be clarified in the text.

4- The main drawback of the manuscript is the lack of detailed insights into the intriguing mechanisms and selectivity.

In Fig. 1, the structure of intermediate int-2 is incorrect. In Lewis acid-catalyzed cycloadditions of BCBs, three possible pathways exist: 1) ring-opening via int-2 as reported by Studer; 2) Similar to Leitch's work, a stepwise ring-opening process of BCB involving a zwitterionic enolate containing a benzylic carbocation moiety (ruled out for only monosubstituted BCBs are suitable substrates); 3) another pathway involves the concerted nucleophilic ring opening of BCBs with a nucleophile. Validating these pathways, regioselectivity ((3+2) versus (4+3)) and diastereoselectivity would necessitate DFT calculations.

5- Did the authors investigate Glorius's BCB (ref. 29) with an acyl pyrazole group in the

(3+2) and (4+3) cycloadditions?

6-Is it possible to convert by-product 4 into the (3+2) cycloadduct in the presence of a Lewis catalyst?

Reviewer #2 (Remarks to the Author):

The submitted manuscript from Ni and Deng describes the addition of silyl enol ethers to bicyclobutanes to generate substituted bicyclohexanes. They achieve this using Lewis acid catalysis, and the reaction is proposed to proceed via a stepwise mechanism. The nucleophilic enol ether attacks the bicyclobutane to generate a new enolate intermediate on the opened cyclobutane, followed by annulation via enolate attack on the O-silyl carbonyl. Using a siloxy-substituted diene, the authors can also achieve formal [4+2] cycloaddition to generate bicyclooctanes (in a manner analogous to a contemporaneous report from Waser and co-workers, cited as the ChemRxiv).

Overall the work is well done, with an extensive scope of silyl enol ethers explored, though a much more limited scope of bicyclobutanes (which is unfortunately the norm in BCB functionalization studies at the moment). Two noteworthy aspects are the scope of enol ethers, and the 2+2 versus 4+2 cycloaddition reactivity depending on substitution. Given the many recent examples of alkene addition to BCBs, this work does stand out with respect to exocyclic heteroatom incorporation for further reactivity (fluorination, elimination, etc. from Fig 5). I think this work could be suitable for Nature Communications, though the conceptual novelty of the approach is undercut somewhat by prior work that is not cited or mentioned (more below).

In terms of significance, this work is another example of alkene + BCB → bicyclohexane, of which there have been many examples in the past 2-3 years (as outlined by the authors in the introduction). Here a different mechanistic approach is taken, relying on a (proposed) 2-electron, stepwise mechanism akin to other Lewis acid catalyzed cycloaddition reactions. This mechanism is supported by the isolation of compound 4, which is the product of nucleophilic attack, but silyl transfer prior to ring closing.

However, the authors fail to cite a key study of bicyclohexane formation via a stepwise mechanism involving initial nucleophilic attack. Only two examples of nucleophilic attack to bicyclobutanes are given in the introduction, both involving indoles (and one from the authors' labs).

Leitch and co-workers have previously demonstrated enolate nucleophiles can add to bicyclobutanes in the same fashion as the mechanism described herein.
Chem. Commun., 2023, 59, 13847-13850.

And Glorius and co-workers recently extended this concept to addition of enolates derived from isocyanides (giving azabicycloheptanes).
Angew. Chem. Int. Ed., 2024, 63, e202402730.

These papers and the precedent established must be included/cited, especially in light of the very similar mechanistic concepts (central cycle in Fig 6, though the Leitch work does not invoke Lewis acid catalysis, and the Glorius work has the cyclobutane nucleophile rebound

to the isocyanide carbon). This seems like a glaring omission given the similarity in concept and mechanism.

Certainly there are also opportunities to contrast the present work with especially the Leitch work, specifically with respect to the reaction scope and ability to access tertiary alcohol derivatives (submitted paper) versus ketones (Leitch work).

In terms of data analysis, conclusions, etc., the work appears sound. All new compounds are characterized appropriately, with good quality NMR spectra included in the SI.

One aspect to be addressed: On page 12, lines 210-213, the authors state: "Considering the proposed reaction process, the origin of distinction in diastereoselectivity possibly arose from the steric bulk of the attached substituent geminal to silyloxyl group on silyl enol ether in the Aldol-type cyclization step." Earlier in the main text, the reader is referred to the Supporting Information for details, but section 5 in the SI (pg 58) simply re-states the observations. The authors should propose a stereochemical model to explain this in more detail (e.g. Felkin-Anh type model?).

In terms of methodology, the general procedure in the SI for silyl enol ether formation must include the scale (or range of scales) of the reactions in mmol and solvent volume. Also, the mass of material produced for each example must be included (not just the percent yield). The general procedures and examples for bicyclohexane formation do include this needed information.

A couple of questions to be addressed:

-Is this reaction scalable beyond 0.2 mmol? An example at 1.0 mmol (e.g. formation of 3a) should be included if possible. If not, this should be noted in the text.

-Can the silyl enol ether be generated in situ? Either immediately before addition of BCB and Lewis acid, or in a one-pot reaction with BCB present at the outset?

A minor point: throughout the authors use square brackets to indicate the type of cycloaddition, specifically [2+3] and [4+3]. These should be rounded brackets, not square brackets. Square bracket for cycloaddition notation indicates the number of electrons involved from each substrate (Woodward-Hoffmann notation), whereas rounded brackets indicate the number of atoms involved. The enolate + BCB reaction is a formal [2+2] cycloaddition / is a formal (2+3) cycloaddition.

Reviewer #3 (Remarks to the Author):

This manuscript presented a Lewis acid catalyzed formal [2+3] cycloaddition between silyl enol ethers and bicyclo[1.1.0]butanes, allowing straightforward access to various bicyclo[2.1.1]hexanes. The reaction is very robust with broad substrate scope from widely available ketones. Notably, this method could be extended to formal [4+3] cycloaddition in the presence of silyl dienol ethers, thus leading to the construction of structurally intriguing bicyclo[4.1.1]octanes. Additionally, this method is synthetically valuable for the divergent synthesis of complex saturated three-dimensional carbocycles, as demonstrated by useful transformations of resulting cycloadducts and rapid modification of natural products through silylation of ketones and formal cycloadditions. Therefore, this work represents a significant advance in this field. The supplementary materials appear to be complete and clear.

Overall, this referee would consider the recommendation for the publication of this manuscript in Nature Communications.

There are some issues to be addressed.

1. Could this reaction be extended to BCBs with esters and amides functional groups?
2. Have the authors examined the silyl enol ethers from esters and aldehydes?
3. Did the authors assess trimethylsilyl enol ether?
4. The author should cite the very recent work on $[2\pi+2\sigma]$ cycloaddition reaction of alkenes and BCBs from Glorius group preprinted on ChemRxiv: 10.26434/chemrxiv-2024-p2pvb.
5. In line 177, "The yield was of isolated and purified products" is repetitive.

Response to Reviewers

(A) Reviewer 1:

1. 1) “The Leitch group pioneered the design of Lewis acid-catalyzed (3+2) cycloaddition of bicyclo[1.1.0]butanes (BCBs) in 2022. The design of Lewis acid-catalysed (3+2) cycloaddition of bicyclo[1.1.0]butanes (BCBs), including the cycloaddition of BCBs with electron-enriched alkenes (ref. 22 and 25), is general, as evidenced by the rapidly increasing number of reports cited by the authors (ref. 27-32). Some (3+2) cycloadditions of BCBs are now even enantioselective (J. Am. Chem. Soc. 2023, 145, 24466–24470; J. Am. Chem. Soc. 2024, 146, 8372–8380). However, the (4+3) cycloadducts, BCOs, are intriguing and significant, as they could serve as novel isosteres for multisubstituted benzenes, as reported by Grygorenko (Chem. Eur. J. 2023, e202303859). The present Lewis acid-catalyzed cycloaddition of BCBs with dienes is a unique (4+3) cycloaddition, offering a catalytic approach to obtain valuable BCOs. Consequently, the aforementioned significant studies should be referenced, the introduction requires revision,”

Our response:

Thanks for the referee 1’s suggestions. We have cited the references on enantioselective (3+2) cycloaddition mentioned by referee 1 as ref. 22-23 and revised the introduction. We added the following description to the first paragraph of page 3 in the revised manuscript “*Subsequently, Bach and coworkers achieved an enantioselective cycloaddition of 2(1H) - quinolones and BCBs with a chiral mediator*²². Recently, Jiang and coworkers developed a highly enantioselective cycloaddition of vinylazaarenes and BCBs under photosensitized chiral phosphoric acid catalysis²³.”

The important work from Grygorenko group was cited as ref. 51 and we added the following sentence “*Grygorenko and coworker analyzed the structural characteristics of BCOs and suggested that they could serve as novel isosteres for multi-substituted*

*benzenes*⁵¹.” to the second paragraph of page 10 in the revised manuscript.

2) “and the substrate scope of the (4+3) reaction needs to be expanded.”

Our response:

The scope of (4+3) reaction is well illustrated, including the silyl dienol ethers bearing 1-, 3-, 4-alkyl group or 3, 4-dialkyl groups. In the revised manuscript, a new case of (4+3) cycloaddition between 3-phenyl silyl dienol ether **1bb** and BCB **2a** was added. The cycloadduct **5e** was obtained in 52% yield. (Fig. 3, compound **5e**), which demonstrated (4+3) cycloaddition could tolerate aromatic substituent.

3) “Furthermore, the authors should cite recent progress in (hetero)-(4+3) cycloadditions of BCBs (such as ChemRxiv, DOI: 10.26434/chemrxiv-2024-r9c2g)”

Our response:

We have cited this paper mentioned by reviewer 1 as ref. 35 (previously published on ChemRxiv, DOI: 10.26434/chemrxiv-2024-r9c2g) which has been published on *Angew. Chem. Int. Ed* (*Angew. Chem., Int. Ed.* **n/a**, e202405222, doi: 10.1002/anie.202405222) in the revised manuscript.

2. 1) “The authors suggested that enolate addition to BCBs is the reaction mechanism. However, this process is not frequently observed in BCB chemistry. The initial attack by the carbanion, is found in recent work from Leitch and co-workers on enolate additions to bicyclobutanes that is not cited (*Chem. Commun.*, 2023, 59, 13847 – 13850).”

Our response:

Our reaction proceeded with nucleophilic addition of silyl enol ether to the Lewis acid-activated bicyclobutanes, which is similar to the Mukaiyama-Michael or aldol reaction mechanistically. On the other hand, what Leitch described is a formal cycloaddition of BCBs via an enolate intermediate that was generated from treatment of glycine imine or arylacetate derivatives with stoichiometric amount of LHMDS. We added following sentence to the introduction of revised manuscript “*Very recently, Leitch group described a formal cycloaddition of BCBs via an enolate intermediate by treatment of glycine imine or arylacetate derivatives with stoichiometric amount of LHMDS*³⁶.” (See first paragraph of page 5 in the revised manuscript.) We have cited the Leitch’s paper as ref. 36.

2) “It is disappointing that the 1,3-disubstituted BCBs cannot be used in the (3+2) and (4+3) cycloadditions of BCBs. Have any other byproducts been observed in the reaction of these 1,3-disubstituted BCBs? This should be clarified in the text.”

Our response:

In our further studies of cycloaddition with 3-substituted BCB **2g** and **2h**, we found that the reaction could proceed smoothly to give the desired cycloadducts **3bh** and **3bi** in 54% and 43% yield. These results have been added to the revised manuscript (Fig. 2) and supporting information (Page 43–44 and 149–150).

We added a description in the first paragraph of page 10 in the revised manuscript “*To our delight, the reaction with 3-substituted BCB **2g** and **2h** also proceeded smoothly to give the cycloadducts bearing three quaternary carbon stereocenters (**3bh** and **3bi**)*”. We attempted the reaction of silyl dienol ether **1az** with 3-phenyl BCB **2h**. However, desired product **5n** was not identified from the reaction mixture. Minor byproduct cyclobutene **S1** isomerized from BCB **2g** was obtained (this result was added to page 51 of revised supplementary information).

4. 1) “The main drawback of the manuscript is the lack of detailed insights into the intriguing mechanisms and selectivity. In Fig. 1, the structure of intermediate int-2 is incorrect.”

Our response:

We thank referee 1 for pointing out this error in drawing, and we revised the structure of **Int 2** in Fig. 1 in the revised manuscript.

2) “In Lewis acid-catalyzed cycloadditions of BCBs, three possible pathways exist: 1) ring-opening via int-2 as reported by Studer; 2) Similar to Leitch’s work, a stepwise ring-opening process of BCB involving a zwitterionic enolate containing a benzylic carbocation moiety (ruled out for only monosubstituted BCBs are suitable substrates); 3) another pathway involves the concerted nucleophilic ring opening of BCBs with a nucleophile. Validating these pathways, regioselectivity ((3+2) versus (4+3)) and diastereoselectivity would necessitate DFT calculations.”

Our response:

As pointed out by referee 1, a stepwise ring-opening process of BCB involving a zwitterionic enolate containing a benzylic carbocation moiety could be ruled out for monosubstituted BCBs.

We believe well designed experimental mechanistic studies would also be very effective to provide evidence for elucidating reaction pathways. For example, if this cycloaddition proceeded by a concerted (3+2) pathway, both alkene carbons of the silyl enol ether would show ^{13}C kinetic isotope effect (KIE) by Singleton’s method (Singleton, D. A.; Thomas, A. A., *J. Am. Chem. Soc.* **1995**, *117*, 9357-9358). On the other hand, if the reaction proceeded by a stepwise mechanism, only one of the two carbons of the silyl enol ether would show a ^{13}C KIE effect. We carried out ^{13}C KIE experiment of cycloaddition reaction between silyl enol ether **1w** and BCB **2a**. The pronounced carbon isotope effect was only observed on the carbon **c** of the silyl enol ether **1w** when the ^{13}C integration ratio of the standard sample was compared to that prepared with excess **1w** ($^{13}\text{C}(\text{standard})/^{13}\text{C}(\text{sample})$ at $\text{C}(\text{c}) = 1.017$, average of three runs). These results are consistent with a stepwise pathway as we proposed. (This ^{13}C KIE experiment has been added on the second paragraph in page 13 and Fig. 6a of the revised manuscript and page 65-66 of the revised supplementary information)

Table 1. ^{13}C integration of the samples of **3w**

1st run

C#	Standard	Sample 1 (excess 1w)	Standard/Sample 1	change (%)
a	1.000	1.000	1.000	0
b	1.009	1.010	0.999	-0.1
c	0.978	0.958	1.021	2.1

2nd run

C#	Standard	Sample 2 (excess 1w)	Standard/Sample 2	change (%)
a	1.000	1.000	1.000	0
b	1.009	1.003	1.005	0.5
c	0.978	0.964	1.015	1.5

3rd run

C#	Standard	Sample 3 (excess 1w)	Standard/Sample 3	change (%)
a	1.000	1.000	1.000	0
b	1.009	1.004	1.005	0.5
c	0.978	0.962	1.017	1.7

Therefore, the formal cycloaddition between silyl enol ether and BCB initiated from the nucleophilic addition of silyl enol ether to Lewis acid activated BCB and followed by the intramolecular Aldol-type reaction. The observation of diastereoselectivity in the cases of **3ao** and **3ap** also provide the evidence of this mechanism.

The regioselectivity of formal (3+2) and (4+3) cycloadditions with silyl dienol ethers appears to be substrate-dependent. As demonstrated by the formation of (3+2) and

(4+3) cycloadducts (**3u**, **3al**, **3am**, **5a-5h**), the steric and electronic properties of the substituent on silyl dienol ethers have a significant influence on which pathway is favored. For example, silyl dienol ethers bearing a 3-alkyl/phenyl or 1-alkyl substituent would favor the formation of (4+3) cycloadduct. The silyl dienol ethers with a less bulky substituent such as methyl or alkenyl on the 4 position would favor the (4+3) cycloaddition pathway (**5g** and **5h**). The reaction of cyclic silyl dienol ether **1al** and BCB **2a** provided (3+2) cycloadduct **3al** exclusively. On the other hand, the (4+3) pathway would lead to the formation of **3al'**, which is much more congested sterically than **3al** (see scheme below). Therefore, the (4+3) pathway is energetically unfavored.

Additionally, the diastereoselectivity is originated from the substituents of trisubstituted silyl enol ether **1ao** and **1ap**. Following referee 2's suggestion, we proposed a Felkin-Anh type model of the zwitterionic intermediate (**Int 5a** and **Int 5b**) to explain the diastereoselectivity (see the scheme below). In the reaction of silyl enol ether **1ao** and BCB **2a** to access **3ao**, a zwitterionic intermediate **Int 5a** was formed after the nucleophilic addition of silyl enol ether to BCB. According to the Felkin-Anh model, the large substituent of the α -carbon of carbonyl compound prefers a perpendicular orientation relative to the carbonyl group, which gave two possible conformations (**Int 5a-1** and **Int 5a-2**). The stronger gauche interaction between the ethyl and phenyl group versus that between the hydrogen and phenyl group rendered **Int 5a-1** a more favorable conformation. **Int 5a-1** led to a high diastereoselectivity in favor of diastereomer **3ao**.

Felkin-Anh type model

In the reaction of silyl enol ether **1ap** and BCB **2a** to access **3ap**, a zwitterionic intermediate **int 5b** was formed after the nucleophilic addition of silyl enol ether to BCB. The steric interaction of methyl and ethyl group is not as pronounced, possibly resulting in low diastereoselectivity (This diastereoselectivity explanation was added on page 60–63 in revised supplementary information).

Therefore, we thought that these experiments evidently support the mechanism and DFT calculation is not necessary.

5. “Did the authors investigate Glorius’s BCB (ref. 29) with an acyl pyrazole group in the (3+2) and (4+3) cycloadditions?”

Our response:

We investigated the formal (3+2) cycloadditions with BCB **2h** bearing acyl pyrazole group. The reaction between silyl enol ether **1a** and BCB bearing an acyl pyrazole group (**2h**) proceeded smoothly to give the corresponding cycloadduct in 43% yield. This result has been added to the revised manuscript (Fig. 2 and first paragraph of page 10) and supporting information (Page 44 and 146). However, the reaction with the BCB bearing an acyl pyrazole group (**2h**) and silyl dienol ether **1at** could not give any desired product.

6. “Is it possible to convert by-product **4** into the (3+2) cycloadduct in the presence of a Lewis catalyst?”

Our response:

We treated the byproduct **4** with Lewis acid, no cycloadduct **3a** was detected. This result could be rationalized by difference in activity of the **Int 5** and byproduct **4** (see scheme below). The formal cycloaddition of silyl enol ether **1a** and BCB **2a** started from a nucleophilic addition of silyl enol ether **1a** to the Lewis acid-activated BCB **2a**, the formed zwitterionic intermediate **Int 5** then underwent an intramolecular Aldol reaction to give the cycloadduct **3a**. For the aldol reaction, the silyl oxonium ion is expected to be more active than Lewis acid-activated carbonyl group as an electrophile while the enolate is expected to be more active than silyl enol ether as a

nucleophile.

(B) Reviewer 2:

1. “Leitch and co-workers have previously demonstrated enolate nucleophiles can add to bicyclobutanes in the same fashion as the mechanism described herein. Chem. Commun., 2023, 59, 13847-13850.”

Our response:

Our reaction proceeded with nucleophilic addition of silyl enol ether to the Lewis acid-activated bicyclobutanes, which is similar to the Mukaiyama-Michael or Aldol reaction mechanistically. On the other hand, what Leitch described is a formal cycloaddition of BCBs via an enolate intermediate that was generated from treatment of glycine imine or arylacetate derivatives with stoichiometric amount of LHMDS. We added following sentence to the introduction of revised manuscript “*Very recently, Leitch group described a formal cycloaddition of BCBs via an enolate intermediate by treatment of glycine imine or arylacetate derivatives with stoichiometric amount of LHMDS³⁶.*” (See first paragraph of page 5 in the revised manuscript.) We have cited the Leitch’s work as ref. 36.

2. “And Glorius and co-workers recently extended this concept to addition of enolates derived from isocyanides (giving azabicycloheptanes). Angew. Chem. Int. Ed., 2024, 63, e202402730.”

Our response:

We have cited the paper in the revised manuscript as ref. 38.

3. "One aspect to be addressed: On page 12, lines 210-213, the authors state: "Considering the proposed reaction process, the origin of distinction in diastereoselectivity possibly arose from the steric bulk of the attached substituent geminal to silyloxy group on silyl enol ether in the Aldol-type cyclization step." Earlier in the main text, the reader is referred to the Supporting Information for details, but section 5 in the SI (pg 58) simply re-states the observations. The authors should propose a stereochemical model to explain this in more detail (e.g. Felkin-Anh type model?)."

Our response:

Following referee 2's suggestion, we proposed a Felkin-Anh type model of the zwitterionic intermediate (**Int 5a** and **Int 5b**) to explain the diastereoselectivity (see scheme above). In the reaction of silyl enol ether **1a0** and BCB **2a** to access **3a0**, a zwitterionic intermediate **Int 5a** was formed after the nucleophilic addition of silyl enol ether to BCB. According to the Felkin-Anh model, the large substituent of the α -carbon of carbonyl compound prefers a perpendicular orientation relative to the carbonyl group, which gave two possible conformations (**Int 5a-1** and **Int 5a-2**). The stronger gauche interaction between the ethyl and phenyl group versus that between the hydrogen and phenyl group rendered **Int 5a-1** a more favorable conformation. **Int 5a-1** led to a high diastereoselectivity in favor of diastereomer **3a0**. In the reaction of silyl enol ether **1ap** and BCB **2a** to access **3ap**, a zwitterionic intermediate **int 5b** was formed after the nucleophilic addition of silyl enol ether to BCB. The steric interaction of methyl and ethyl group is not as pronounced, possibly resulting in low diastereoselectivity (This diastereoselectivity explanation was added on page 60–63 in revised supplementary information).

Felkin-Anh type model

4. “In terms of methodology, the general procedure in the SI for silyl enol ether formation must include the scale (or range of scales) of the reactions in mmol and solvent volume. Also, the mass of material produced for each example must be included (not just the percent yield). The general procedures and examples for bicyclohexane formation do include this needed information.”

Our response:

Thank you for referee 2’s suggestion. We have added the mass of starting material of the reactions to the general procedure of the revised supporting information (page 4-10). The amount of the product in mass (mg or g) was added for each product.

5. “A couple of questions to be addressed: -Is this reaction scalable beyond 0.2 mmol?

An example at 1.0 mmol (e.g. formation of 3a) should be included if possible. If not, this should be noted in the text.”

Our response:

We conducted a preparative scale reaction at 2.0 mmol scale, which afforded cycloadduct **3w** in 91% yield (770 mg, see Fig. 4a of the revised manuscript).

6. “-Can the silyl enol ether be generated in situ? Either immediately before addition of BCB and Lewis acid, or in a one-pot reaction with BCB present at the outset?”

Our response:

We tried the reaction with acetophenone and BCB **2a** in the presence of TIPSOTf and no desired product was observed. In addition, further attempt with Yb(OTf)₃ and TIPSOTf led to a complex mixture.

7. “A minor point: throughout the authors use square brackets to indicate the type of cycloaddition, specifically [2+3] and [4+3]. These should be rounded brackets, not square brackets. Square bracket for cycloaddition notation indicates the number of electrons involved from each substrate (Woodward-Hoffmann notation), whereas rounded brackets indicate the number of atoms involved. The enolate + BCB reaction is a formal [2+2] cycloaddition / is a formal (2+3) cycloaddition.”

Our response:

Following referee 2's suggestion, we changed all the square brackets to rounded bracket where we describe the type of cycloaddition in the revised manuscript.

(C) Reviewer 3:

1. "Could this reaction be extended to BCBs with esters and amides functional groups?"

Our response:

In the reaction of BCB bearing ester and amide, we could not observe any reactivity (see the unsuccessful substrates in the page 12 of the revised supplementary information). To our delight, the reaction with BCB **2h** bearing an acyl pyrazole group could work well to give the cycloadduct **3bi** in 43% yield. This acyl pyrazole group could be readily transformed to ester or amide as illustrated in the literature (*Angew. Chem., Int. Ed.* **2023**, *62*, e202305043). This example has been added to the revised manuscript (Fig. 2) and supporting information (Page 43–44 and 150).

2. "Have the authors examined the silyl enol ethers from esters and aldehydes?"

Our response:

We examined some silyl enol ethers derived from δ -valerolactone or phenylpropyl aldehyde. However, these substrates decomposed under our standard reaction condition. (see the unsuccessful substrates in the page 12 of the revised

supplementary information)

3. “Did the authors assess trimethylsilyl enol ether?”

Our response:

Trimethylsilyl enol ether decomposed under our standard reaction condition. (see the unsuccessful substrates in the page 12 of revised supporting information)

4. “The author should cite the very recent work on $[2\pi+2\sigma]$ cycloaddition reaction of alkenes and BCBs from Glorius group preprinted on ChemRxiv: 10.26434/chemrxiv-2024-p2pvb.”

Our response:

We have cited this paper mentioned by reviewer 3 as ref. 29 (previously published on ChemRxiv, DOI: 10.26434/chemrxiv-2024-p2pvb) which has been published on J. Am. Chem. Soc. (*J. Am. Chem. Soc.*, doi:10.1021/jacs.4c04403) in the revised manuscript. We changed the sentence “*Lately, Glorius accomplished the coupling of phenol and BCB by leveraging a photoredox process (Fig. 1b–1)*²⁶.” to “*Lately, Glorius accomplished the coupling of phenol and BCB by leveraging a photoredox process, which have been applied in the formal cycloaddition of non-activated alkenes (Fig. 1b–1)*²⁸⁻²⁹.” in first paragraph of page 3 in the revised manuscript.

5. “In line 177, “The yield was of isolated and purified products” is repetitive.”

Our response:

Thank you for the referee 3 for pointing out this error. We have removed the repetitive sentence in the revised manuscript.

In addition to these revisions, we have made the following minor revisions.

1. We changed the title “*Facile Access to Bicyclo[2.1.1]hexanes by Formal Cycloaddition of Silyl Enol Ethers and Bicyclo[1.1.0]butanes with Lewis Acids*” to “*Facile access to bicyclo[2.1.1]hexanes by Lewis acid-catalyzed formal cycloaddition between silyl enol ethers and bicyclo[1.1.0]butanes*” following the word number limitation of title.
2. We added the single crystal structure of **5a** (CCDC 2358618) in the revised manuscript and supplementary information (Table 4, page 70).

REVIEWERS' COMMENTS

Reviewer #1 (Remarks to the Author):

The minor comments I made previously have been satisfactorily resolved. The inclusion of ^{13}C kinetic isotope effect (KIE) and the Felkin-Anh type model is excellent. In terms of the BCB substrate scope, in addition to monosubstituted BCBs, 1,3-disubstituted BCBs also yield the desired (3+2) cycloadducts smoothly. Therefore, the structure of 2 in Fig.2 should be redrawn to demonstrate the generalizability of BCB scopes. Great work!
I would be happy to see this manuscript be published in Nature Communications.

Reviewer #2 (Remarks to the Author):

The authors have carried out extensive revisions of their manuscript to address all reviewer comments. I support publication of the revised version.

Reviewer #3 (Remarks to the Author):

The authors have fully revised or rebutted to the concerns from the previous reviewers. I think the work is ready for publication.

Response to reviewers

Reviewer 1

“In terms of the BCB substrate scope, in addition to monosubstituted BCBs, 1,3-disubstituted BCBs also yield the desired (3+2) cycloadducts smoothly. Therefore, the structure of **2** in Fig.2 should be redrawn to demonstrate the generalizability of BCB scopes.”

Our response:

We thank the reviewer 1's suggestion. We redraw the structure of BCB **2** in Fig. 2 to represent all the BCB substrate scope.